# Clustering ICU patients with sepsis based on the patterns of their circulating biomarkers: A secondary analysis of the CAPTAIN prospective multicenter cohort study

Benoît Misset[1,2]*, François Philippart[3,4], Catherine Fitting[5], Jean-Pierre Bedos[6], Jean-Luc Diehl[7,8], Olfa Hamzaoui[9], Djillali Annane[10,11], Didier Journois[12], Marianna Parlato[5], Virginie Moucadel[13], Jean-Marc Cavaillon[5], Joël Coste[14,15], for the CAPTAIN Study Group[¶]

1 Intensive Care Department, Liège University Hospital, Liège, Belgium, 2 Infection, Immunity and Inflammation Research Unit, GIGA I3, Liège University, Liège, Belgium, 3 Service de Médecine Intensive et Réanimation, Groupe Hospitalier Paris Saint-Joseph, Paris, France, 4 Unité Endotoxines, Structures et Réponse de l'Hôte, Département de Microbiologie, Institut de Biologie Intégrative de la Cellule, Paris Saclay, Saclay, France, 5 Unit Cytokines & Inflammation, Institut Pasteur, Paris, France, 6 Service de Réanimation, Hôpital André Mignot, Versailles, France, 7 Service de Réanimation Médicale, Hôpital Européen Georges Pompidou, Assistance Publique—Hôpitaux de Paris, Paris, France, 8 INSERM UMR S1140, Université Paris Descartes, Paris Sorbonne Cité, Paris, France, 9 Service de Réanimation, Hôpital Antoine Béclère, Assistance Publique—Hôpitaux de Paris, Clamart, France, 10 Service de Réanimation, Hôpital Raymond Poincaré, Assistance Publique—Hôpitaux de Paris, Garches, France, 11 Université Versailles Saint-Quentin, Versailles, France, 12 Service de Réanimation Chirurgicale, Hôpital Européen Georges Pompidou, Assistance Publique—Hôpitaux de Paris, Paris, France, 13 BioMérieux SA, Lyon, France, 14 Unité de Biostatistiques et d'Epidémiologie, Hôpital Cochin, Assistance Publique—Hôpitaux de Paris, Paris, France, 15 Université de Paris, Paris, France

¶ Members of the Combined Approach for The eArly diagnosis of INfection in sepsis (CAPTAIN) study group is provided in the Acknowledgments (collaborators section).
* benoit.misset@chuliege.be

**Data Availability Statement:** All relevant data are within the paper and its Supporting Information file.

## Abstract

### Background

Although sepsis is a life-threatening condition, its heterogeneous presentation likely explains the negative results of most trials on adjunctive therapy. This study in patients with sepsis aimed to identify subgroups with similar immune profiles and their clinical and outcome correlates.

### Methods

A secondary analysis used data of a prospective multicenter cohort that included patients with early assessment of sepsis. They were described using Predisposition, Insult, Response, Organ failure sepsis (PIRO) staging system. Thirty-eight circulating biomarkers (27 proteins, 11 mRNAs) were assessed at sepsis diagnosis, and their patterns were determined through principal component analysis (PCA). Hierarchical clustering was used to group the patients and $k$-means algorithm was applied to assess the internal validity of the clusters.

**Funding:** Grant from the Programme Hospitalier de Recherche Clinique of the French Ministry of Health (PHRC AOM 09143) and from the Institut Mérieux-Institut Pasteur collaborative research partnership. The study was sponsored by the Département de la Recherche Clinique et du Développement de l'Assistance Publique-Hôpitaux de Paris. This project was part of Advanced Diagnostics for New Therapeutic Approaches, a program dedicated to personalized medicine, coordinated by Mérieux Alliance and supported by the French public agency, OSEO. These funding sources had no role in the design of this study, analyses, interpretation of the data, or decision to submit the results.

**Competing interests:** I have read the journal's policy and the authors of this manuscript have the following competing interests: Virginie Moucadel, PhD, is employed by bioMérieux SA, a private company specialized in in vitro diagnostics. The authors declare no other potential conflicts of interest in relation with the subject of the present manuscript.

## Results

Two hundred and three patients were assessed, of median age 64.5 [52.0–77.0] years and SAPS2 score 55 [49–61] points. Five main patterns of biomarkers and six clusters of patients (including 42%, 21%, 17%, 9%, 5% and 5% of the patients) were evidenced. Clusters were distinguished according to the certainty of the causal infection, inflammation, use of organ support, pro- and anti-inflammatory activity, and adaptive profile markers.

## Conclusions

In this cohort of patients with suspected sepsis, we individualized clusters which may be described with criteria used to stage sepsis. As these clusters are based on the patterns of circulating biomarkers, whether they might help to predict treatment responsiveness should be addressed in further studies.

## Trial registration

The CAPTAIN study was registered on clinicaltrials.gov on June 22, 2011, # NCT01378169.

## Introduction

Sepsis is a clinical picture of organ dysfunctions elicited by an infection, and associated with immune dysregulation [1]. Its mortality varies between 25 and 60% [2, 3]. The intensity of the organ dysfunctions are usually assessed by the SOFA score [4]. Immune dysregulation is complex and not fully deciphered [5], follows from the activation by both pathogen and danger-associated molecular patterns [6], and is associated with a variety of immune pathways including inflammation, compensatory anti-inflammation, and low adaptive profile [7, 8]. It is likely dependent on underlying diseases [9], genetic predisposition [10] and the causal agent of infection [11]. All these dimensions of sepsis are included in the Predisposition, Insult, Response, Organ failure sepsis (PIRO) classification system, a tool proposed in 2001 to characterize and stage sepsis [12]. As numerous trials failed to improve unselected cohorts of patients with sepsis [13, 14], "endotypes" describing patient groups with similar genetic, epigenetic or proteomic pattern, have been proposed [15]. They are excepted to help to predict treatment responsiveness and not just differences in prognosis [16]. Endotyping aims at categorizing the different pathways involved [14, 16] in order to select patients as potential targets of specific treatments [17, 18]. Data on endotypes are scarce because the collection of many biomarkers is not available in daily routine [19]. To comply with the recommendations of the surviving sepsis campaign [3], sepsis must be suspected and treated before the infection is confirmed, and biomarkers might help to differentiate patients with bacterial infection from those with other causes of immune dysregulation [20]. A better understanding of subgroups within the heterogeneous host response to infection is important both for a better understanding of the biology of sepsis but also for the next generation of trials of more precise interventions for sepsis.

In a multicenter prospective cohort called CAPTAIN that included patients with suspected sepsis for whom circulating proteins or mRNAs from circulating leukocytes were assessed, these biomarkers were not able to discriminate patients with *versus* without a documented causal infection [21]. Then, we hypothesized that a clustering approach may help defining subgroups of similar patients in multidimensional populations.

In the present study, using the same cohort of patients with sepsis, we aimed at identifying homogeneous subgroups in terms of circulating biomarkers, and clinical phenotypes and mortality correlates.

## Methods

### Study design

This study is a secondary analysis of the observational multicenter prospective CAPTAIN study (Combined Approach for The eArly diagnosis of INfection in sepsis) [21]. It was designed and conducted according to STROBE (STrengthening the Reporting of OBservational studies in Epidemiology) guidelines (see Table A in S1 File) [22].

### Ethics and study registration

The protocol was approved by the "Comité de Protection des Personnes Ile de France XI" (#2010-A00908-31-10056) on September 13, 2010 and registered on clinicaltrials.gov (NCT01378169) https://clinicaltrials.gov/ct2/show/NCT01378169. According to French national regulations, written consent of the patients was required but waived for the unarousable ones, and obtained if the study still required specific samples when the patient awoke.

### Setting

Patients were recruited from December 2011 to April 2013 in seven ICUs from five hospitals in Paris area.

### Participants

Eligible ICU patients were those patients with suspected sepsis. The inclusion criteria were hypothermia (below 36.0˚C) or hyperthermia (over 38.0˚C), and at least one criterion of systemic inflammatory response syndrome (SIRS) [12] as soon as the physician considered antibiotic therapy. Other inclusion criteria were age over 18 years, no treatment limitation and no obvious immunosuppression.

Demographics, reasons for ICU admission, underlying diseases, simplified acute severity score (SAPS 2) [20], physiological data, Sequential Organ Failure Assessment (SOFA) score [21] and length of organ failure support were collected at admission to the ICU, at inclusion in the study and over the ICU stay. The population characteristics have been published previously [21]. Briefly, 363 patients were screened and 279 included. Based on the data obtained in the 72 hours after inclusion, infection could not be ascertained in one third of the patients after adjudication by two investigators who were blind to the biomarkers. The biomarkers were found to discriminate poorly between patients with *versus* without a documented causal infection [21].

The present analysis focused on those patients who were still in ICU after day 3, because we wanted to describe their clinical phenotypes including the "certainty", and not the "suspicion", of infection in the phenotypical criteria of the potential clusters. The rationale to describe clusters of phenotypes was to confirm (or not) that different patterns of endotypes—which are intended to describe pathophysiological pathways—were associated with specific phenotypes. As phenotypes were outcomes of our research, we considered that 72 hours was necessary to distinguish suspicion and confirmation of the infection. We excluded those patients having not at least one available value for all the 38 biomarkers collected in the first two days of inclusion because principal component analysis (see below) does not handle with missing data, leading to 203 patients available for analysis. All included patients had an increase of at least

two points of SOFA score in the previous 48 hours and fulfilled the characteristics of the Sepsis-3 definition [1] despite being included before its publication.

## Biomarkers assessment

We collected whole blood samples at day 0 and 1 of inclusion to assess 38 biomarkers, reported as potential indicators of infection or mortality during sepsis [23]. The techniques of assessment are described in the S1 File. For biomarkers whose value was below the lower limit of quantification (LLoQ), we attributed a value of LLoQ / $\sqrt{2}$. For biomarkers whose value was over the upper limit of quantification (ULoQ), we attributed the ULoQ value. The description of the 1) kits for soluble markers concentration measure, 2) lower and upper limits of quantification for each plasma biomarker, 3) primer and probe designs for mRNA biomarkers, and 4) distribution of missing values and determination of the cut-off to create binary variables have been provided previously [21].

## Clinical phenotypes description

We described the patients clinical phenotypes according to the PIRO classification system [12, 24] where predispositions (P) related with chronic status and disease, insult (I) with the cause of the suspected sepsis, response (R) with clinical response to this cause, and organ dysfunction (O) with the nature and severity of the organ dysfunctions. P items were age, gender, body mass index (BMI), Mac Cabe score, chronic lung, cardiac, renal or hepatic insufficiencies, diabetes or malignancy. I items were bacterial infection of the lung, abdomen or urinary tract. R items were body temperature, blood lymphocyte and platelet counts, prothrombin time and serum lactates as indicators of inflammatory response, coagulation activation and tissue hypoxia; pneumonia or bacteremia occurring after day 5 of the ICU stay as indicators of immune dysfunction. O items were sequential organ failure assessment (SOFA) and each of its sub-components (respiratory, nervous, cardiovascular, liver, coagulation and kidney dysfunctions) within two days of inclusion. Outcome was defined as the mortality at the end of the ICU stay.

Infections were confirmed *a posteriori*, based on criteria which confirm infection as much as possible, either with or without positive cultures. The definition of infection and its causal link with organ dysfunction required medical interpretation [25] and were based on IDSA guidelines [26]. They were adjudicated blindly to the studied biomarkers, by two investigators (FP and BM). They reviewed the patients" records, including clinical history, results of routine morphologic, biological, or microbiological tests, and response to therapies during the days following inclusion. Strains were considered as infecting, colonizing or contaminants. Infection could be considered as present despite the absence of a positive microbiological sample, for example in cases of abscess or pneumonia [26]. When bacteremia was present, it was linked to most probable anatomical focus of infection. Viruses were only searched in case of Influenza suspicion and were classified as non-septic SIRS. Disagreements on classification were resolved after discussion between the two adjudicators.

## Statistical analysis

The determination of classes in numerical taxonomy is generally achieved by cluster analysis of a resemblance matrix, which is a combination of similarities (or distances) between all pairs of objects, *e.g.* patient's biomarkers. Here, such a simple process appeared inadequate because of the large and heterogeneous scales of biomarkers involved. Transformation and reduction of data were necessary to obtain a homogeneous scale of independent data. After adequate transformation of the data, we therefore performed factor analysis by principal component

analysis (PCA) and further used several standardized factor scores for each individual as input (in the resemblance matrix) to the clustering method.

## Data transformation

For all considered biomarkers, we determined the maximum value of the blood levels (Cmax) obtained at day 0 and day 1 of inclusion. We chose peak values because we were unable to ascertain the precise date of sepsis onset, due to the variability of the syndrome and the high number of biomarkers used, leading to missing values in several biomarkers. These values were natural log-transformed (ln(*marker*+0.0001) to normalize their distribution (convert the skewed distribution of these variables to approximate normality) and further standardized to have 0 mean and unit variance.

## Principal component analysis

To evidence patterns, transformed Cmax of biomarkers were used to build correlation matrices (Pearson coefficients), which were then studied by PCA, followed by varimax rotations of retained components. PCA is a statistical procedure that summarizes the information content in large data tables by means of a smaller set of "summary indices" that can be more easily visualized and analyzed. PCA is considered the reference method to identify the unobservable, "latent" factors or dimensions that *underlie* or *structure* a set of observed variables. The patterns obtained were uncorrelated linear combinations of normalized and standardized biomarkers, and sorted by decreasing variance of rates explained, whose coefficients, the "loadings", are interpretable as correlation coefficients between patterns and original biomarkers. These loadings help identify the "nature" or "meaning" of the patterns: "loadings" > 0.40 are usually considered to indicate substantial *correlation*. The number of components-patterns to retain was determined by the Horn and Velicer methods as recommended [27]. These components define the dimensionality of the reduced space and correspond to the underlying latent factors or patterns. The remaining components (not retained) represent the residual variability (measurement error, single marker unrelated to the others).

## Clustering

The scores of the patients on the components-patterns retained were selected for cluster analysis. Hierarchical cluster analysis (Ward method) was used to obtain the initial cluster grouping because of the lack of *a priori* knowledge of the number of clusters involved. The number of clusters selected was based on standard statistical criteria (optimal values of R2, pseudo-F, pseudo-t2 and cubic clustering criterion, which all reflect some balance between within and between cluster variances), as recommended [28].

## Cluster internal validity

Two methods were used to evaluate the stability and the replicability of the hierarchical cluster solution; (1) a *k*-means algorithm: this method does not assume a hierarchical relationship among clusters and allows for relocation of cases throughout the clustering process (reducing the risk of misassignment common to hierarchical cluster method [28]); (2) a subsample analysis: the hierarchical cluster analysis was repeated with a random 50% sample of the initial population to investigate whether subjects clustered similarly when they were distributed in subsamples [29].

### Description of the clusters

The clusters obtained were finally compared for each individual circulating biomarker and for the clinical variables usually used to describe patients with sepsis. These variables were classified according to the different categories of the PIRO system to facilitate reading and interpretation. For each phenotype qualitatively described, we defined two groups of patients according to its presence or absence. We compared the proportions of patients in each cluster with these phenotypes using non-parametric Fisher exact test. For each phenotype defined with a quantitative value, we assessed correlations of the phenotype with each cluster using a Spearman rank test and we compared the values of each phenotype within each cluster using Kruskall-Wallis statistics. The quantitative values are displayed as median [Q1-Q3] and the qualitative values as n (%). We considered a p-value below 0.05 for statistical significance.

SAS 9.4 package was used for all analysis (SAS Inst., Cary, NC, USA).

## Results

### Patients

Out of 363 patients screened for biomarkers, 279 patients were included in the CAPTAIN cohort, but 33 died or were discharged from the ICU before day 3, leading to 246 eligible patients. Of these, 43 had a least one missing value among the 38 biomarkers, leading to 203 evaluable patients (Fig 1). The population characteristics according to the PIRO system, are reported in Table 1. Their median age was 64.5 [52.-77.0] years, median SAPS II score, 55 [49–61] points, and they were included 23 [11–45] hours after ICU. Among them, 189 (77%) were in the ">5-year life expectancy" category of the Mac Cabe score, 116 (47%) had underlying diseases, the suspected infection was confirmed for 171 (70%). After day 5 of ICU, 26 (13%) had acquired pneumonia and 8 (4%) bacteremia. Mechanical ventilation, vasopressors, renal replacement therapy and low-dose steroids were used in 182 (74%), 95 (38%), 19 (8%) and 22 (9%) patients, respectively, and 58 (29%) patients died in the ICU. The levels of each single biomarker for the total cohort have been published previously [21]. The description of the patients discharged or died before day-3 are displayed in Table B in S1 File.

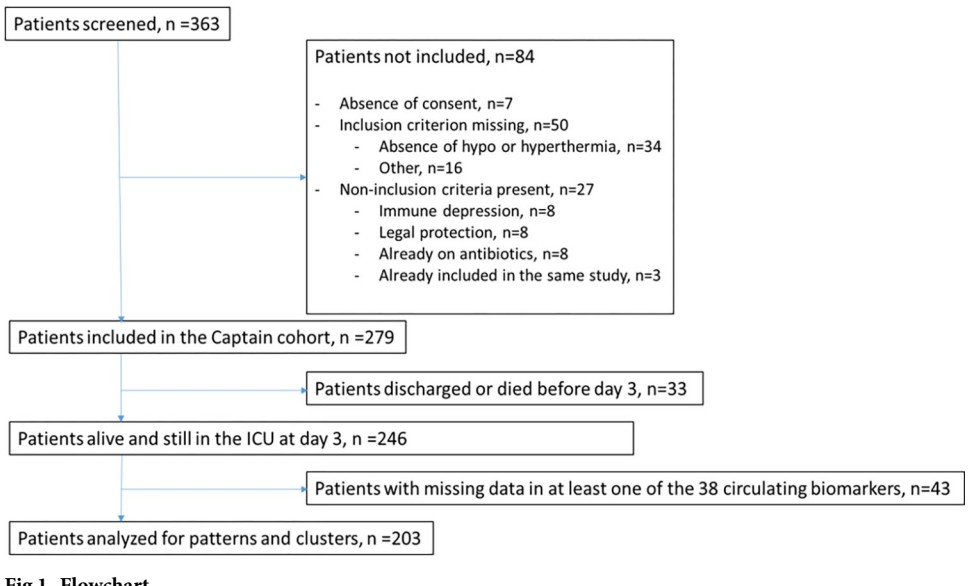

**Fig 1. Flowchart.**

**Table 1. Characteristics of the patients, organized according to the PIRO system and outcome.**

| PIRO category | | Variable | n (%) or med [Q1-Q3] |
|---|---|---|---|
| *Predisposition* | | Age (years) | 64.5 [52.0–77.0] |
| | | Male sex | 160 (65) |
| | | BMI (kg/m$^2$) | 25.7 [21.6–30.0] |
| | | Mc Cabe score, % prediction > 5 years | 189 (77) |
| | | COPD | 46 (19) |
| | | Cardiac insufficiency | 25 (10) |
| | | Diabetes | 53 (22) |
| | | Chronic renal insufficiency | 22 (9) |
| | | Solid tumor | 34 (14) |
| | | Hematologic malignancy | 4 (2) |
| | | Chronic hepatic insufficiency | 18 (7) |
| | | Any prior disease | 115 (47) |
| *Insult* | At inclusion | Infection due to Gram positive bacteria | 76 (31) |
| | | Infection due to Gram negative bacteria | 124 (50) |
| | | Pneumonia | 123 (50) |
| | | Intra-abdominal infection | 14 (6) |
| | | Urinary tract infection | 19 (8) |
| | | Confirmed infection | 171 (70) |
| *Response* | At inclusion | Temperature (˚C) | 38.2 [37.5–38.8] |
| | | Lymphocyte count (/mm$^3$) | 905 [640–1390] |
| | | Blood platelets (10$^3$/mm$^3$) | 187 [134–268] |
| | | Prothrombin time (%) | 66 [59–82] |
| | | Blood lactates (meq/L) | 1.6 [1.0–2.3] |
| | | PaO2 (mmHg) | 88 [73–145] |
| | | FiO2 (%) | 40 [30–60] |
| | | PaC02 (mmHg) | 39 [34–46] |
| | | Serum creatinin (μmol/L) | 92 [71–182] |
| | | Blood hematocrit (%) | 32.2 [28.5–38.6] |
| | | White blood cell count (/mm$^3$) | 13,200 [9,480–18,600] |
| | | Respiratory rate (/min) | 26 [22–33] |
| | | Heart rate (/min) | 106 [95–125] |
| | | Mean arterial pressure (mmHg) | 73 [63–94] |
| | | Urinary output (L/24h) | 1.30 [0.81–2.00] |
| | | SAPS II score (points) | 55 [49–61] |
| | After ICU day 5 | ICU acquired pneumonia | 26 (13) |
| | | ICU acquired bacteremia | 8 (4) |
| *Organ failure* | At inclusion | Total SOFA score (points) | 6 [3–9] |
| | | Respiratory SOFA score (points) | 2 [0–3] |
| | | Neurological SOFA score (points) | 0 [0–2] |
| | | Circulatory SOFA score (points) | 0 [0–1] |
| | | Hepatic SOFA score (points) | 0 [0–0] |
| | | Coagulation SOFA score (points) | 0 [0–1] |
| | | Kidney SOFA score (points) | 0 [0–2] |
| | During the ICU stay | Mechanical ventilation | 182 (74) |
| | | Non invasive ventilation | 15 (6) |
| | | Vaso-active drugs | 95 (38) |
| | | Renal replacement therapy | 19 (8) |
| | | Low doses steroid therapy | 22 (9) |

## Patterns of circulating biomarkers and clusters of patients

PCA of the 38 biomarkers provided five main components-patterns, which explained 30.7%, 9.7%, 8.0%, 5.5% and 4.7% (total 58.6%) of the variance, respectively. The biomarkers with a high loading (|loading| > 0.40) are displayed in Table C in S1 File. Pattern #1 gathers circulating biomarkers and mRNAs linked to both pro- and anti-inflammatory response and to altered immunity. Pattern #2 gathers only circulating biomarkers associated with pro- and anti-inflammatory response and shares numerous cytokines and chemokines with pattern #1 (i.e., IL-1Ra, IL-6, IL-8; MCP-1, G-CSF, GM-CSF, and MIP-1β). Pattern #3 gathers only mRNA markers linked to both pro- and anti-inflammatory response. Pattern #4 displays specific biomarkers not shared with any other clusters (i.e., Galectin-9, SuPAR, MIF, and Ferritin). Similarly, pattern #5 has its specific biomarkers (RANTES, sTREM-1). These 5 patterns allowed to build 6 clusters of patients with homogeneous biological profiles. The Table D in S1 File displays the scores of the 5 patterns within each cluster of patients. The use of a *k*-means algorithm (non-hierarchical method), with the number of clusters set to 6, led to similar clustering as with the Ward method, with satisfactory agreement (Carmer's V = 0.63). Similar clustering solutions were found with analysis of a random 50% of the sample (Cramer's V = 0.59). These results support the robustness of the six clusters.

The Table 2 shows the criteria of the PIRO profile of sepsis. "Predisposition" items of the PIRO system differed moderately across clusters. Among the "insult", "response", and "organ failure" categories, the most different items between clusters were infection certainty, blood lactate levels, serum creatinine levels, urinary output, survival, circulatory and renal SOFA sub-scores, and use of renal replacement therapy, vaso-pressors and steroids. The Table 3 shows that the levels of the 38 individual circulating biomarkers differed markedly across the clusters and that biomarkers of the same category (pro-inflammatory, anti-inflammatory, adaptive immunity) displayed consistent values within each cluster. In these tables, the color code (from dark red to high level, to dark blue for low level) illustrates the differences within each item of the clusters. Based on these comparisons, six clusters can be distinguished according to the certainty level of the causal infection, the existence of inflammation, use of renal and/or hemodynamic support, pro-inflammatory and anti-inflammatory activity, and markers of adaptive profile. Four clusters (clusters #2, #3, #4 and #6) were associated with high mortality (> 30%) and a low adaptive profile (Table 2), and represented 53% of the cohort (Fig 2). Clusters #2 and #6 exhibited both high levels of inflammatory and anti-inflammatory mediators, but differed with regards to CRP and ferritin, #3 displayed anti-inflammatory mediators at low level, and #4 displayed both inflammatory and anti-inflammatory mediators at low level (Fig 2).

## Discussion

In a prospective cohort of ICU patients suspected of sepsis, through the levels of circulating biomarkers indicative of pro-inflammation, anti-inflammation or adaptive immunity and the use of unsupervised statistical approaches, we individualized six different clusters of patients with homogeneous profiles regarding sepsis clinical staging. These clusters presented with different immune and clinical profiles, making them potential targets for individualized therapies.

Sepsis is a life-threatening condition elicited by various infectious conditions, with a heterogeneous presentation, and an outcome impacted by both the pathogen and host characteristics [1]. This phenotypic polymorphism led to the proposal of the PIRO classification and staging system in 2001 to help individualize future therapies [12]. A better understanding of subgroups within the heterogenous host response to infection is important both for a better

**Table 2. Value of each clinical criterion of the PIRO profile of sepsis and outcome in each cluster.**

| PIRO category | Variable | n | unit | Cluster 1 86 | | Cluster 2 43 | | Cluster 3 34 | | Cluster 4 18 | | Cluster 5 11 | | Cluster 6 11 | | p value |
|---|---|---|---|---|---|---|---|---|---|---|---|---|---|---|---|---|
| | | | | med or % | [Q1-Q3] | med or % | [Q1-Q3] | med or % | [Q1-Q3] | med or % | [Q1-Q3] | med or % | [Q1-Q3] | med or % | [Q1-Q3] | |
| **Predisposition** | | | | | | | | | | | | | | | | |
| | Age | | years | 64 | [52–74] | 63 | [52–77] | 70 | [58–78] | 54 | [38–69] | 74 | [49–84] | 77 | [63–87] | 0.02 |
| | BMI | | $kg/m^2$ | 26.1 | [21.4–31.4] | 24.8 | [20.9–27.8] | 26.3 | [23.9–30.4] | 24.5 | [20.2–24.5] | 27.2 | [22.8–37.2] | 24.6 | [22.7–28.8] | 0.35 |
| | Male sex | | % | 62% | | 77% | | 67% | | 83% | | 45% | | 64% | | 0.18 |
| | Mc Cabe score (% > 5 years) | | % | 72% | | 84% | | 73% | | 83% | | 91% | | 64% | | 0.19 |
| | COPD | | % | 24% | | 28% | | 21% | | 6% | | 9% | | 9% | | 0.36 |
| | Cardiac insufficiency | | % | 9% | | 12% | | 18% | | 6% | | 9% | | 0% | | 0.66 |
| | Diabetes | | % | 20% | | 12% | | 29% | | 17% | | 28% | | 28% | | 0.42 |
| | Chronic renal insufficiency | | % | 5% | | 5% | | 26% | | 0% | | 0% | | 27% | | 0.001 |
| | Solid tumor | | % | 15% | | 12% | | 18% | | 6% | | 18% | | 18% | | 0.82 |
| | Hematologic malignancy | | % | 15% | | 20% | | 17% | | 0% | | 0% | | 0% | | 0.83 |
| | Chronic hepatic insufficiency | | % | 10% | | 2% | | 9% | | 11% | | 0% | | 9% | | 0.51 |
| | Any prior disease | | % | 56% | | 46% | | 79% | | 39% | | 54% | | 64% | | 0.03 |
| **Insult** | | | | | | | | | | | | | | | | |
| at inclusion | GPC infection | | % | 7% | | 16% | | 12% | | 11% | | 9% | | 9% | | 0.65 |
| | GNB infection | | % | 37% | | 65% | | 65% | | 67% | | 54% | | 45% | | 0.01 |
| | Pneumonia | | % | 44% | | 56% | | 56% | | 67% | | 45% | | 18% | | 0.2 |
| | Intra-abdominal infection | | % | 2% | | 12% | | 3% | | 11% | | 9% | | 9% | | 0.12 |
| | UTI | | % | 3% | | 12% | | 18% | | 0% | | 0% | | 27% | | 0.01 |
| | Confirmed infection as a cause | | % | 53% | | 88% | | 85% | | 83% | | 64% | | 54% | | 0.0001 |
| **Response** | | | | | | | | | | | | | | | | |
| at inclusion | Temperature | | °C | 38.2 | [37.5–38.7] | 37.9 | [37.4–38.7] | 38.3 | [37.8–38.9] | 39.0 | [38.0–39.3] | 38.6 | [38.2–40.0] | 38.2 | [37.5–38.9] | 0.04 |
| | Lymphocyte count | | $/mm^3$ | 960 | [690–1,540] | 820 | [500–1,190] | 910 | [525–1,200] | 990 | [860–1,200] | 700 | [540–1,390] | 1,085 | [685–1,695] | 0.29 |
| | Blood platelets | | $10^3/mm^3$ | 170 | [129–252] | 171 | [131–215] | 222 | [145–306] | 252 | [203–379] | 225 | [145–293] | 95 | [72–156] | 0.008 |
| | Prothrombin time | | % | 71 | [62–84] | 69 | [57–80] | 69 | [58–78] | 74 | [61–82] | 83 | [76–87] | 35 | [28–40] | 0.001 |
| | Blood lactates | | meq/L | 1.35 | [0.90–1.80] | 2,00 | [1.40–2.70] | 1.55 | [1.35–2.80] | 1.35 | [0.90–2.30] | 1.45 | [0.80–2.30] | 2.9 | [2.30–4.50] | 0.0002 |
| | PaO2 | | mmHg | 94 | [74–123] | 79 | [62–123] | 85 | [74–106] | 86 | [77–100] | 71 | [57–125] | 95 | [81–168] | 0.34 |
| | FiO2 | | % | 35 | [30–50] | 47 | [30–80] | 40 | [30–55] | 40 | [30–50] | 30 | [24–39] | 50 | [40–70] | 0.05 |
| | PaCO2 | | mmHg | 41 | [34–47] | 40 | [34–48] | 39 | [34–49] | 39 | [34–42] | 31 | [27–36] | 36 | [31–47] | 0.03 |
| | Serum creatinin | | µmol/L | 90 | [70–150] | 120 | [75–195] | 195 | [90–280] | 85 | [80–140] | 140 | [80–220] | 260 | [220–520] | 0.0007 |
| | Blood hematocrit | | % | 34.2 | [29.0–39.4] | 35.7 | [29.9–40.0] | 29.6 | [27.2–33.0] | 28.4 | [25.1–30.4] | 32.8 | [28.5–38.1] | 31.2 | [29.2–38.5] | 0.001 |
| | White blood cell count | | $/mm^3$ | 12,380 | [8,700–15,450] | 13,450 | [10,220–18,700] | 14,350 | [11,300–19,300] | 15,600 | [9,200–18,800] | 13,900 | [8,200–21,700] | 14,000 | [10,140–40,300] | 0.32 |
| | Respiratory rate | | /min | 25 | [21–31] | 25 | [22–30] | 29 | [23–33] | 30 | [27–33] | 37 | [28–40] | 26 | [21–34] | 0.006 |
| | Heart rate | | /min | 100 | [91–111] | 112 | [98–128] | 115 | [98–139] | 112 | [102–121] | 111 | [99–116] | 110 | [98–148] | 0.03 |

(Continued)

**Table 2.** (Continued)

| PIRO category | Variable | unit | Cluster 1 (86) med or % | [Q1-Q3] | Cluster 2 (43) med or % | [Q1-Q3] | Cluster 3 (34) med or % | [Q1-Q3] | Cluster 4 (18) med or % | [Q1-Q3] | Cluster 5 (11) med or % | [Q1-Q3] | Cluster 6 (11) med or % | [Q1-Q3] | |
|---|---|---|---|---|---|---|---|---|---|---|---|---|---|---|---|
| | Mean arterial pressure | mmHg | 75 | [67–89] | 67 | [60–97] | 71 | [59–98] | 82 | [70–101] | 81 | [66–113] | 59 | [46–65] | 0.009 |
| | Urinary output | L/24h | 1,200 | [700–1,700] | 1,442 | [930–1,990] | 1,075 | [400–2,075] | 1,900 | [1,500–2,275] | 2,125 | [1,750–2,300] | 85 | [5–400] | 0.0001 |
| | SAPS II score | points | 55 | [49–61] | 52 | [49–61] | 55 | [49–60] | 51 | [50–57] | 53 | [44–59] | 59 | [56–61] | 0.34 |
| over the ICU stay | acquired pneumonia | % | 8% | | 14% | | 12% | | 22% | | 9% | | 9% | | 0.59 |
| | acquired bacteremia | % | 2% | | 2% | | 3% | | 6% | | 9% | | 18% | | 0.12 |
| **Organ failure** | | | | | | | | | | | | | | | |
| over the ICU stay | Total SOFA score | points | 5 | [2–5] | 6 | [4–10] | 5 | [4–9] | 3 | [2–7] | 7 | [4–8] | 11 | [10–14] | 0.0001 |
| | Respiratory SOFA score | points | 2 | [1–3] | 2 | [2–3] | 2 | [1–3] | 2 | [2–3] | 2 | [1–3] | 2 | [2–3] | 0.2 |
| | Neurological SOFA score | points | 0 | [0–1] | 0 | [0–3] | 0 | [0–1] | 0 | [0–0] | 1 | [0–4] | 0 | [0–2] | 0.24 |
| | Circulatory SOFA score | points | 0 | [0–0] | 0 | [0–3] | 0 | [0–3] | 0 | [0–0] | 0 | [0–0] | 4 | [1–4] | 0.0001 |
| | Hepatic SOFA score | points | 0 | [0–0] | 0 | [0–0] | 0 | [0–0] | 0 | [0–0] | 0 | [0–2] | 2 | [0–2] | 0.13 |
| | Coagulation SOFA score | points | 0 | [0–1] | 0 | [0–1] | 0 | [0–0] | 0 | [0–0] | 0 | [0–1] | 1 | [3–4] | 0.1 |
| | Kidney SOFA score | points | 0 | [0–1] | 0 | [0–2] | 1 | [1–2] | 0 | [0–0] | 1 | [1–4] | 4 | [3–4] | 0.0001 |
| | Mechanical ventilation | % | 65% | | 86% | | 76% | | 94% | | 54% | | 78% | | 0.02 |
| | Non invasive ventilation | % | 7% | | 2% | | 3% | | 0% | | 27% | | 11% | | 0.06 |
| | Vaso-active drugs | % | 30% | | 58% | | 47% | | 12% | | 9% | | 73% | | 0.0001 |
| | Renal replacement therapy | % | 2% | | 2% | | 24% | | 0% | | 0% | | 45% | | 0.0001 |
| | Low doses steroid therapy | % | 1% | | 19% | | 15% | | 0% | | 9% | | 27% | | 0.0005 |
| **Outcome** | | | | | | | | | | | | | | | |
| | Death at day-14 | % | 5% | | 14% | | 12% | | 0% | | 0% | | 37% | | 0.0001 |
| | Death at day-30 | % | 10% | | 19% | | 21% | | 33% | | 9% | | 73% | | 0.0001 |
| | Death at day-60 | % | 13% | | 28% | | 29% | | 44% | | 19% | | 82% | | 0.0001 |

The clinical criteria are sorted according to the PIRO classification system. In each raw, the dark red color indicates the highest value and dark blue color indicates the lowest value, for those variables which are the most significantly different across the clusters.

**Table 3. Value of each circulating biomarker of sepsis in each cluster.**

| | | Cluster 1 | | Cluster 2 | | Cluster 3 | | Cluster 4 | | Cluster 5 | | Cluster 6 | | |
|---|---|---|---|---|---|---|---|---|---|---|---|---|---|---|
| **Biomarker category** | | n = 86 | | n = 43 | | n = 34 | | n = 18 | | n = 11 | | n = 11 | | |
| *Variable* | *unit* | med | [Q1-Q3] | med | [Q1-Q3] | med | [Q1-Q3] | med | [Q1-Q3] | med | [Q1-Q3] | med | [Q1-Q3] | *p value* |
| *Inflammatory mediators or biomarkers* | | | | | | | | | | | | | | |
| *Cytokines* | | | | | | | | | | | | | | |
| *TNF-α* | ng/L | 10 | [10–10] | 10 | [10–10] | 10 | [10–10] | 10 | [10–10] | 10 | [10–10] | 10 | [10–220] | 0.0001 |
| *TNF-α RNA* | CNRQ | 0.015 | [0.012–0.018] | 0.017 | [0.011–0.022] | 0.013 | [0.007–0.019] | 0.008 | [0.006–0.010] | 0.018 | [0.014–0.023] | 0.009 | [0.004–0.016] | 0.0001 |
| *IL-1β RNA* | CNRQ | 0,008 | [0.006–0.011] | 0.008 | [0.005–0.008] | 0.006 | [0.005–0.013] | 0.004 | [0.003–0.005] | 0.013 | [0.009–0.018] | 0.004 | [0.002–0.009] | 0.0001 |
| *IL-18* | ng/L | 45.2 | [16.1–80.7] | 53.8 | [35.9–97.4] | 77.4 | [38.7–197.1] | 46.6 | [29.2–72.3] | 123.5 | [123.5–368.0] | 365.0 | [333.7–924.3] | 0.0001 |
| *IL-15* | ng/L | 2.5 | [2.50–2.50] | 2.5 | [2.50–2.50] | 2.5 | [2.50–2.50] | 2.5 | [2.50–2.50] | 48.4 | [2.5–90.9] | 25.7 | [6.7–107.4] | 0.0001 |
| *IL-6* | ng/L | 45 | [15–149] | 1,298 | [327–5,344] | 119 | [44–342] | 59 | [24–485] | 873 | [193–2107] | 8,333 | [1,259–85,509] | 0.0001 |
| *GM-CSF* | ng/L | 9.8 | [4.0–28.9] | 609.1 | [64.0–1,754.9] | 13.5 | [4.0–53.9] | 4.0 | [4.0–43.2] | 18.2 | [4.0–575.8] | 411.5 | [293.0–13,027.0] | 0.0001 |
| *Chemokines and receptors* | | | | | | | | | | | | | | |
| *MCP-1* | ng/L | 49 | [5–85] | 173 | [98–927] | 54 | [5–84] | 75 | [44–192] | 240 | [48–379] | 342 | [127–11,415] | 0.0001 |
| *MIF* | μg/L | 8.5 | [4.4–14.2] | 5.1 | [2.7–7.5] | 15.5 | [6.7–22.2] | 11.7 | [7.7–23.9] | 12.5 | [8.7–24.3] | 42.0 | [23.3–48.6] | 0.0001 |
| *Rantes CCL5* | μg/L | 14.2 | [9.6–14.2] | 10.4 | [7.3–16.3] | 12.1 | [8.1–15.7] | 15.8 | [10.4–19.1] | 57.6 | [22.4–115.1] | 6.6 | [4.1–31.6] | 0.0001 |
| *IP-10* | ng/L | 165 | [59–363] | 284 | [117–552] | 505 | [216–904] | 282 | [185–800] | 6,034 | [1,631–21,478] | 836 | [580–60,981] | 0.0001 |
| *IL-8* | ng/L | 12.5 | [4.0–23.8] | 73.4 | [31.6–337.2] | 26.0 | [213.5–47.9] | 14.3 | [9.4–26.0] | 24.4 | [4.0–137.0] | 494.0 | [187.0–1,494.0] | 0.0001 |
| *MIP-1β* | ng/L | 39 | [20–60] | 88 | [40–243] | 62 | [24–132] | 53 | [37–73] | 305 | [70–450] | 149 | [119–1,229] | 0.0001 |
| *CX3CR1 RNA* | CNRQ | 14.1 | [10.9–18.9] | 7.2 | [4.0–12.5] | 9.5 | [4.7–16.0] | 5.7 | [4.1–7.0] | 10.4 | [6.8–16.4] | 4.2 | [1.7–7.6] | 0.0001 |
| *Others* | | | | | | | | | | | | | | |
| *C reactive protein* | mg/L | 146 | [68–203] | 296 | [241–331] | 265 | [194–346] | 235 | [154–318] | 310 | [265–454] | 208 | [153–249] | 0.0001 |
| *Procalcitonin* | μg/L | 1.3 | [1.2–1.3] | 2.7 | [1.4–6.3] | 2.2 | [1.6–4.5] | 1.8 | [1.5–2.2] | 1.6 | [1.4–5.3] | 7.2 | [2.8–18.9] | 0.0001 |
| *SuPAR* | ng/L | 6.14 | [4.35–8.34] | 7.27 | [5.41–11.49] | 15.33 | [12.89–18.27] | 11.58 | [6.4–14.2] | 10.4 | [7.0–14.1] | 20.6 | [11.8–35.0] | 0.0001 |
| *Visfatin* | μg/L | 5.56 | [4.52–7.00] | 4.98 | [4.22–5.85] | 5.47 | [4.42–6.75] | 4.45 | [4.07–5.34] | 5.17 | [4.50–6.03] | 38.46 | [10.31–140.11] | 0.0001 |
| *PSP* | ng/L | 69 | [45–139] | 325 | [139–613] | 341 | [145–1,121] | 144 | [111–321] | 115 | [66–257] | 780 | [349–1,538] | 0.0001 |
| *sB7-H6* | ng/L | 24.6 | [16.5–27.7] | 27.5 | [18.3–31.4] | 23.9 | [15.9–29.1] | 19.2 | [11.2–26.5] | 28.2 | [17.7–35.8] | 0.0 | [0.0–0.0] | 0.0004 |
| *MMP-8* | μg/L | 20 | [10–45] | 77 | [47–167] | 85 | [30–192] | 46 | [20–110] | 74 | [64–197] | 198 | [93–212] | 0.0001 |
| *sTREM-1* | μg/L | 2.2 | [1.3–4.0] | 4.1 | [2.0–5.5] | 4.7 | [3.1–6.5] | 1.0 | [0.6–2.6] | 2.3 | [1.4–4.3] | 4.3 | [3.3–8.6] | 0.0001 |
| *HMGB1 RNA* | CNRQ | 3.64 | [3.14–4.22] | 3.58 | [3.03–4.05] | 3.89 | [3.19–3.89] | 3.22 | [2.47–3.61] | 3.35 | [2.53–4.03] | 3.81 | [3.00–4.45] | 0.03 |
| *Ferritin* | μg/L | 1 | [1–2] | 2 | [1–2] | 211 | [117–434] | 248 | [165–501] | 273 | [178–516] | 960 | [732–1,075] | 0.0001 |
| *Galectin 9* | μg/L | 5.7 | [3.4–7.4] | 5.8 | [3.2–8.5] | 12.5 | [9.4–17.6] | 8.2 | [5.0–12.8] | 8.1 | [6.6–12.5] | 25.7 | [19.2–35.9] | 0.0001 |
| *S100A9 RNA* | CNRQ | 15.4 | [11.4–15.4] | 28.2 | [21.7–36.5] | 24.1 | [17.0–28.3] | 14.6 | [9.7–20.3] | 24.3 | [14.1–32.7] | 30.2 | [18.9–37.4] | 0.0001 |
| *Anti-inflammatory mediators* | | | | | | | | | | | | | | |
| *IL-1Ra* | CNRQ | 15.0 | [15.0–15.0] | 18.3 | [15.0–620.4] | 15.0 | [15.0–16.7] | 15.0 | [15.0–15.0] | 215.4 | [15.0–632.3] | 3,174.8 | [927.0–5,062.0] | 0.0001 |
| *IL-10* | ng/L | 4.0 | [4.0–4.0] | 4.0 | [4.0–10.9] | 4.0 | [4.0–4.0] | 4.0 | [4.0–4.0] | 4.0 | [4.0–4.0] | 107.9 | [51.3–321.7] | 0.0001 |
| *IL-10 RNA* | CNRQ | 0.17 | [0.12–0.24] | 0.60 | [0.41–1.13] | 0.23 | [0.12–0.40] | 0.09 | [0.07–0.12] | 0.41 | [0.27–0.78] | 0.37 | [0.19–0.57] | 0.0001 |
| *Adaptive immunity* | | | | | | | | | | | | | | |

(Continued)

**Table 3.** (Continued)

| Biomarker category | | Cluster 1 | | Cluster 2 | | Cluster 3 | | Cluster 4 | | Cluster 5 | | Cluster 6 | | |
|---|---|---|---|---|---|---|---|---|---|---|---|---|---|---|
| | | n = 86 | | n = 43 | | n = 34 | | n = 18 | | n = 11 | | n = 11 | | |
| *Variable* | *unit* | med | [Q1-Q3] | med | [Q1-Q3] | med | [Q1-Q3] | med | [Q1-Q3] | med | [Q1-Q3] | med | [Q1-Q3] | *p value* |
| *HLA-DR RNA* | *CNRQ* | 0.34 | [0.27–0.42] | 0.15 | [0.10–0.20] | 0.24 | [0.11–0.41] | 0.18 | [0.15–0.21] | 0.29 | [0.27–0.42] | 0.28 | [0.04–0.36] | *0.0001* |
| *CD74 RNA* | *CNRQ* | 0.59 | [0.46–0.71] | 0.28 | [0.18–0.40] | 0.36 | [0.19–0.63] | 0.27 | [0.27–0.32] | 0.59 | [0.45–0.72] | 0.26 | [0.08–0.53] | *0.0001* |
| *LILRB2 RNA* | *CNRQ* | 0.83 | [0.67–0.99] | 0.93 | [0.82–1.28] | 0.76 | [0.68–0.94] | 0.42 | [0.35–0.54] | 1.12 | [0.71–1.32] | 0.79 | [0.48–1.09] | *0.0001* |
| *CD3 RNA* | *CNRQ* | 1.18 | [0.87–1.51] | 0.56 | [0.34–0.74] | 0.49 | [0.33–1.06] | 0.56 | [0.44–1.12] | 0.79 | [0.49–1.02] | 0.21 | [0.20–0.53] | *0.0001* |
| *Pathogen associated molecular patterns* | | | | | | | | | | | | | | |
| *Peptidoglycan* | *µg/L* | 2.19 | [0.75–3.31] | 2.79 | [1.63–3.79] | 2.63 | [1.75–3.90] | 2.64 | [2.01–3.53] | 4.44 | [2.76–6.08] | 1.89 | [0.75–2.71] | *0.001* |

The individual biomarkers are sorted according to their role in inflammation, anti-inflammation or adaptive immune profile. In each raw, the dark red color indicates the highest value and dark blue color indicates the lowest value. CNRQ = Calibrated Normalized Relative Quantity.

understanding of the biology of sepsis but also for the next generation of trials of more precise interventions for sepsis.

Given the heterogeneity of both clinical and immune presentation of sepsis and the multiple failures of trials in unselected populations [13], a cluster approach has been used by several authors [30]. In these studies, clustering was based either on phenotypes to describe different clinical profiles without addressing immune mechanisms of sepsis, and mostly provide

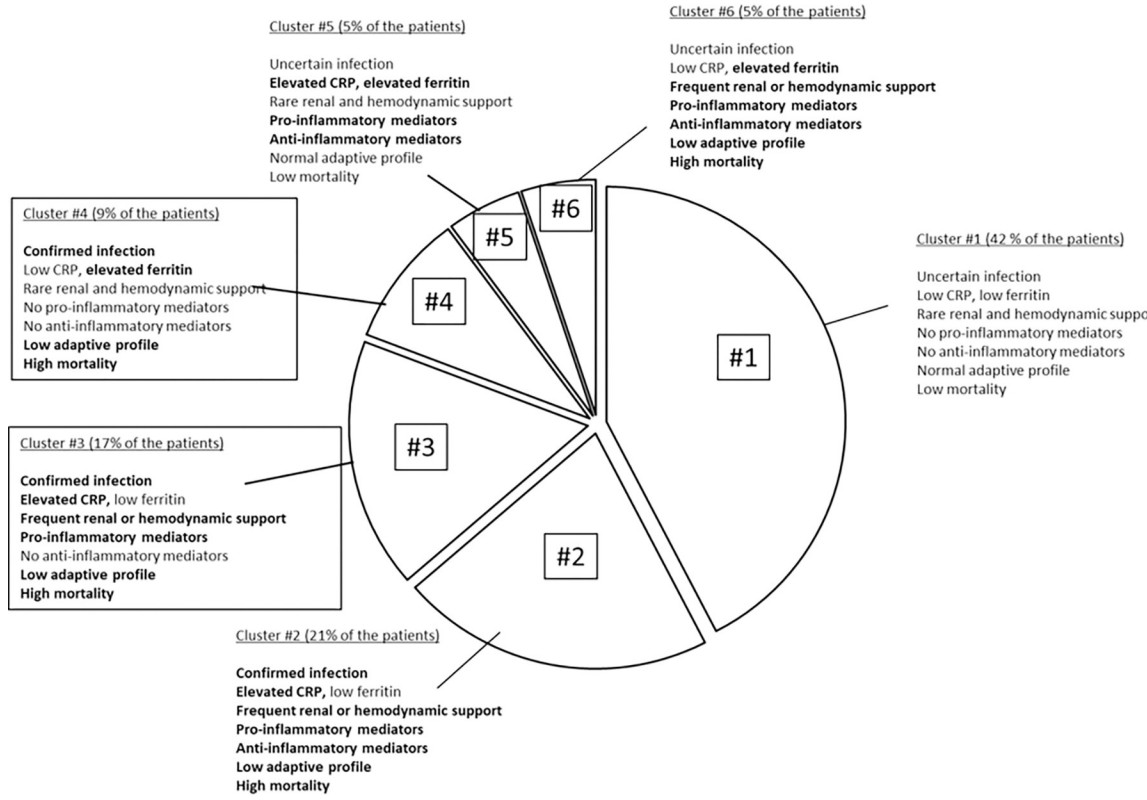

**Fig 2. Distribution and description of the clusters in the cohort.** All the clusters display different characteristics in terms of parameters of the PIRO system, of circulating biomarkers and outcome.

differences in prognosis, or on endotypes, deriving subclasses from genome-wide expression profiling [30]. The latter option, as endotypes are markers of pathophysiological pathways, may help to predict treatment responsiveness [16]. Wong *et al.* identified three pediatric septic shock subclasses named A, B and C [31]. Subclass A displayed a depressed expression of adaptive immune system, of glucocorticoid signaling and zinc-related biology and associated with higher severity and mortality. Scicluna *et al.* identified four sepsis subclasses named Mars 1 to 4 [15]. Subclass Mars1, with a higher mortality, displayed reduced expression of genes involved in innate and immune functions. Subclass Mars3, with a better survival, displayed increased expression of adaptive immune or T-cell functions. Davenport *et al.* identified two sepsis subclasses named SRS1 and SRS2 [32], with SRS1 characterized by a higher mortality and decreased expression of immune functions. Finally, based on gene activation profiles, Sweeney *et al.* identified three sepsis subclasses named inflammopathic, adaptive and coagulopathic [33]. The adaptive subgroup was associated with lower severity and mortality, and the coagulopathic subgroup with higher mortality and clinical coagulopathy. In our study, we also derived clusters from endotypes, but these were assessed with biomarkers made of molecules previously documented to play a role the pathophysiology of sepsis. We used unsupervised statistical approaches to set up clusters, because they explore data without *a priori* classification [34]: principal component analysis, to determine different patterns, hierarchical clustering to group the patients and *k*-means algorithm to assess the internal validity of the clusters.

In our cohort, two clusters (#1 and #5) had a low level of organ dysfunction and mortality. Clusters #2 and #3 displayed high level of infection certainty and inflammation, and differed by their anti-inflammatory status, consistent with the concept of compensatory anti-inflammation and its heterogeneity [8]. Cluster #4 displayed a low level of innate response despite high severity and high ferritin levels. Lastly, two clusters, #5 and #6, were associated with very specific phenotypes, one (#5) with pro and anti-inflammatory high-level profile despite low mortality, and the second one (#6) with high immune alteration and particularly high level of ferritin.

Half our cohort belongs to clusters associated with a high mortality rate. They are characterized by their low adaptive profile at sepsis diagnosis. They differ between themselves by their respective levels of certainty of the causal infection, of CRP and ferritin levels, of renal and hemodynamic level of support, and of pro-inflammatory and anti-inflammatory activities. Each of these characteristics may be available at bedside in parallel of the assessment of infection, organ dysfunction, pro- (for example TNFα and/or IL-18) and anti-inflammatory (for example IL-1Ra and/or IL-10) cytokines, and markers of adaptive function (for example HLA-DR). These criteria are relatively simple and should be validated in external cohorts before they can be used as inclusion criteria in prospective trials.

To use our research to enroll patients in trials, we expect that investigators should select among the phenotypical clusters the one(s) which is(are) of interest for their research, and then include patients according to their endotype pattern at the time of inclusion, that is measure several of the biomarkers that were used in our work. For example, based on Fig 2, clusters 2, 3, and 4 would be good candidates to assess the effects of antimicrobials and/or molecules that restore the immune paralysis (interferon gamma, interleukin-7. . .); cluster 3 would be a good candidate for anti-inflammatory drugs (steroids. . .) or antibodies (anti IL-6. . .); cluster 6 could be a good candidate to test restoring the immune profile (IFN, IL-7. . .). Then, based on Table 3, the investigator could select among the inflammatory mediators (TNF alpha, IL-6. . .), the anti-inflammatory mediators (IL-1Ra, IL-10. . .), and/or the immune adaptive profile (HLA-DR, CD74. . .) to establish inclusion criteria in their trial.

One cannot ascertain the infection before day 2 or 3 in most patients. Therefore, only biomarkers present when the first symptoms of inflammation and the suspicion of sepsis occur

may be useful to select patients in future trials, or to adjust causal (anti-bacterial) therapy of sepsis. Consequently, we consider that only the endotypes (and not the phenotypes) that we found may be useful for this purpose. In future use of these endotypes, a selection of 2 or 3 biomarkers for each immune pathway would be sufficient, limiting the risk of missing values. Future external validation of our results could be based on a minimal dataset based on both the endotypes and phenotypes we found. This data set could be the following: confirmed infection at day 3: yes/no; blood values of CRP and ferritin; use of RRT and/or vaso-pressive drugs; blood levels of TNF-α, IL-6, and IL-8 (pro-inflammatory); blood levels of IL-1Ra and IL-10 (anti-inflammatory); HLA-DR and CD74 (adaptive profile).

Our study has limitations. First, several biomarkers, including some recently described, were not assessed in this cohort, and should be assessed in similar conditions. Especially, biomarkers more specific of endothelial dysfunction or coagulation activation were underrepresented in our panel. Second, although at risk of sepsis, several categories of patients were not included in the cohort, particularly those with prior immune suppression, whose innate and adaptive responses are likely different from the immunocompetent patients. These patients should be investigated using a similar approach. Third, the limitations of the statistical methods should be borne in mind. It is useful to recall that the patterns and clusters that emerge from factorial and taxonomic methods do not exactly correspond to clear-cut groups or endotypes. Despite following recommendations for optimizing method implementation and enhancing reliability of results, emerged clusters may be polluted by misclassification of statistical nature and their meaning require careful analysis. It is therefore crucial to check the stability of the clusters obtained, especially of the smaller ones. Finally, while the generalizability of our study may be reinforced by its multicenter design and the use of internal validity assessments, we did not perform external validation in a separate cohort. This is particularly important for the groups with small numbers of patients in our cohort.

## Conclusion

In a prospective cohort of ICU patients with suspected sepsis, we individualized clusters of patients which may be described with criteria commonly used to stage sepsis in routine practice. As these clusters are based on the patterns of circulating biomarkers, whether they might help to predict treatment responsiveness should be addressed in further studies.

## Supporting information

**S1 File. Methods.** Techniques of biomarkers measurements. Table A. STROBE—Checklist of items that should be included in reports of *cohort studies*. Table B. Characteristics of the patients that were discharged or died before day3 (n = 33). Table C. Circulating biomarkers with a loading > 0.40 or < -0.40 in each main independent patterns obtained after principal component analysis. Table D. Value of each pattern of biomarkers in the identified clusters (med [Q1-Q3]).
(PDF)

## Acknowledgments

We sincerely thank Drs Anne-Françoise Rousseau, Bernard Lambermont and François Jouret for their critical review, Céline Féger, MD (EMIBiotech) and Séverine Marck for their editorial support for this manuscript.

## Collaborators

Members of the Combined Approach for The eArly diagnosis of INfection in sepsis (CAP-TAIN) study group:

Sébastien JACQMIN, Didier JOURNOIS, Alix LAGRANGE, Gabrielle PINOT de VILLE-CHENON (*Hôpital Européen Georges Pompidou, AP–HP—Université Paris Descartes, Service d'Anesthésie-Réanimation)*, Nadia AISSAOUI, Jean-Luc DIEHL, Emmanuel GUEROT, Marion VENOT (*Hôpital Européen Georges Pompidou, AP–HP—Université Paris Descartes, Service de Réanimation Médicale)*, Olfa HAMZAOUI, Dominique PRAT, Benjamin SZTRYMF (*Hôpital Antoine Béclère, AP–HP—Université Paris Sud, Clamart)*, Djillali ANNANE, Virginie MAXIME, Andrea POLITO (*Hôpital Raymond Poincaré, AP–HP—Université Paris Ile de France Ouest, Service de Réanimation Médico-chirurgicale)*, Belaïd BOUHEMAD, Cédric BRUEL, Frédéric ETHUIN, Julien FOURNIER, Maïté GARROUSTE-ORGEAS, Charles GREGOIRE, Nicolas LAU, Benoît MISSET, François PHILIPPART (*Groupe Hospitalier Paris Saint Joseph, Service de Réanimation)*, Jean-Pierre BEDOS, Pierrick CROSNIER, Virginie LAURENT, Sybille MERCERON (*Hôpital André Mignot, Versailles, Service de Réanimation médico-chirurgicale)*, Elsa BOURNAUD, Laurence LECOMTE, Jean-Marc TRELUYER, (*Hôpital Cochin, AP–HP—Université Paris Descartes, Unité de Recherche Clinique)*, Alexandre PACHOT, Javier YUGUEROS-MARCOS, Laurent ESTEVE, Sophie BLEIN, Virginie MOUCADEL (bioMérieux, Lyon & Grenoble), Myriam BEN BOUTIEB, Alexandra ROUQUETTE, Joël COSTE (*Hôpital Cochin, AP–HP—Université de Paris, Unité de Biostatistique et d'Epidémiologie)*, Minou ADIB-CONGUY, Jean-Marc CAVAILLON, Catherine FITTING, Marianna PARLATO, Virginie PUCHOIS, Fernando SOUZA-FONSECA-GUIMARAES (Institut Pasteur, Paris, Unit Cytokines & Inflammation).

The corresponding author of the CAPTAIN study group is Benoît MISSET, MD (benoit.misset@chuliege.be).

## Author Contributions

**Conceptualization:** Benoît Misset, François Philippart, Jean-Marc Cavaillon, Joël Coste.

**Data curation:** Benoît Misset, François Philippart, Catherine Fitting, Jean-Pierre Bedos, Jean-Luc Diehl, Olfa Hamzaoui, Djillali Annane, Didier Journois, Marianna Parlato, Virginie Moucadel.

**Formal analysis:** Benoît Misset, François Philippart, Marianna Parlato, Virginie Moucadel, Jean-Marc Cavaillon, Joël Coste.

**Funding acquisition:** Benoît Misset, Virginie Moucadel, Jean-Marc Cavaillon.

**Investigation:** Benoît Misset.

**Methodology:** Benoît Misset, Joël Coste.

**Project administration:** Catherine Fitting.

**Resources:** Benoît Misset.

**Supervision:** Benoît Misset, François Philippart, Jean-Marc Cavaillon, Joël Coste.

**Validation:** Benoît Misset.

**Visualization:** Benoît Misset.

**Writing – original draft:** Benoît Misset, François Philippart, Jean-Marc Cavaillon, Joël Coste.

**Writing – review & editing:** Benoît Misset, François Philippart, Djillali Annane, Virginie Moucadel, Jean-Marc Cavaillon, Joël Coste.

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
