## [Decision Letter · Decision Letter 0]

23 Aug 2022

PONE-D-22-10420Clustering ICU patients with sepsis based on the patterns of their circulating biomarkers: a secondary analysis of the CAPTAIN prospective multicenter cohort study.PLOS ONE

Dear Dr. Misset,

Thank you for submitting your manuscript to PLOS ONE. After careful consideration, we feel that it has merit but does not fully meet PLOS ONE’s publication criteria as it currently stands. Therefore, we invite you to submit a revised version of the manuscript that addresses the points raised during the review process.

The authors have identified an extremely important area of inquiry relating to the study of patients with sepsis. While this paper has substantial limitations (most of which are acknowledged by the authors), I think that overall it is well done and addresses a critical knowledge gap in the sepsis literature. As such, I think that publication of these findings is important. However, there are some areas that need to be addressed prior publication.

In addition to the reviewer comments, please consider the following suggestions when preparing revisions:

-i think reviewer #1 makes an important point about patients being in the ICU for 3 days—while this may allow for more clear delineation of phenotypes, by day 3 it may be too late to alter treatment, etc, and it seems like it would be quite difficult to wait to enroll patients in a trial until 3 days in ICU

-line 153: can you more clearly define what “recent increase in total SOFA score” means?

-there is a great deal of technical detail in the paper, a much of which I think can be moved to a supplement. I say this as I think the more readable/less technical the main paper is, the more clinicians will read it. Of course, the technical details are important, but I do think that focusing on the findings and the overall implications of such lines of investigation are the most important aspects of this paper. For example, I think that most of the paragraph “biomarkers assessment” can be moved to supplement, with the exception of the first sentence, and the description of how values above and below the limits of detection of each assay were handled.

-can you comment on the utility of inclusion of components from PCA that explain such a small amount of overall variance based on PCA? Perhaps the more general question would be can you detail the reasoning behind how you selected the number of components?

-to me, one of the most important aspects of this analysis is the clinical characteristics of each cluster, but these are not well-defined. For example, many of the characteristics in table 2 are poorly defined: what do “cardiac insufficiency” and “hepatic insufficiency” mean? In terms of malignancy, there is likely a great deal of heterogeneity amongst different tumor types and stages and whether or not the patients are getting treated. “Any prior disease” seems exceedingly broad. Taken together, this makes it nearly impossible to understand how similar/different the clusters are from a clinical standpoint which greatly hampers the interpretation and application of the results, which is a stated goal of the paper.

-the above-mentioned issues also raise some questions about the reported outcome associations for each cluster. Broadly speaking, without a very clear understanding of the clinical characteristics of each cohort, it is difficult to understand if and how much the biomarker profiles aid in outcome prediction s compared to clinical characteristics alone. For example, a patient with stage 4 lung cancer, end-stage HFrEF, and cirrhosis is likely going to have a worse outcome than a patient without those factors and thus, one could argue that the biomarker profiles are not useful in this scenario. While overall I do believe that the biomarker profiles will be useful, and I recognize that this is an extreme example, I think that this a reasonable criticism that will raised by opponents to such an approach as as such I think that addressing it will strengthen the manuscript.

We look forward to receiving your revised manuscript.

Kind regards,

Robert R Ehrman, MD, MS

Academic Editor

PLOS ONE

Journal Requirements:

I have read the journal's policy and the authors of this manuscript have the following competing interests: Virginie Moucadel, PhD, is employed by bioMérieux SA, a private company specialized in in vitro diagnostics. The authors declare no other potential conflicts of interest in relation with the subject of the present manuscript. 

4. One of the noted authors is a group or consortium Captain Study Group. In addition to naming the author group, please list the individual authors and affiliations within this group in the acknowledgments section of your manuscript. Please also indicate clearly a lead author for this group along with a contact email address.

Reviewers' comments:

Reviewer's Responses to Questions

**Comments to the Author**

1. Is the manuscript technically sound, and do the data support the conclusions?

Reviewer #1: Yes

Reviewer #2: Yes

2. Has the statistical analysis been performed appropriately and rigorously? 

Reviewer #1: I Don't Know

Reviewer #2: Yes

3. Have the authors made all data underlying the findings in their manuscript fully available?

Reviewer #1: Yes

Reviewer #2: Yes

4. Is the manuscript presented in an intelligible fashion and written in standard English?

Reviewer #1: Yes

Reviewer #2: Yes

5. Review Comments to the Author

Reviewer #1: The authors propose an unbiased systems biology approach (PCA) to characterize the various sepsis phenotypes based on biomarkers of inflammation (pro and anti), and representing specific immune responses (e.g., IL-18 for inflammasome). The rationale being to better characterize sepsis phenotypes to guide future clinical trials (as per the abstract) and to elucidate disease mechanisms (as per the Introduction). Clinical characterization was based on PIRO, whereas biological characterization was based on an array of 38 selected different proteins and mRNA transcripts. The analysis yielded 6 distinct sub-groups of sepsis, some with higher mortality, and with distinct molecular markers. There are question remaining:

Concerns:

1. Is PCA the best statistical approach? The reliance on a complete dataset leads to the exclusion of many sepsis cases, including perhaps the most import one (those dying within 3 days).

2. Although 33 patients died and were therefore excluded, it would be interesting to describe this special sub-group based on the available, albeit limited data.

3. Most of the molecules used to characterize sepsis biology are reasonable. Certain common biomarkers associated clinically with sepsis, such as PMN% or blast% were not included. Please explain.

4. Why was "viral sepsis" not included? What about fungal sepsis?

5. It is said the the data was "transformed" and "reduced" prior to establishing clusters. Please provide more detail such that the approach could be reproduced by others.

6. Figure 2 is excellent and nicely summarizes the results. However, it would interesting to consider the minimal data set that could be used for future external validation of these sepsis sub-types.

7. From a practical standpoint it is most useful to establish sepsis phenotypes as early as possible (e.g., 3 days would be too long for an interventional clinical trail). As such, it may be more interesting to know what the initial features of each sub-type are rather than "peak values" of biomarkers measured over 24 hrs.

Reviewer #2: This study was interesting in clustering sepsis profiles and analyzing them across various clinical and management factors and bio markers. As this was a secondary data analysis, not sure why a significant portion fo the methods section was used to describe the CAPTAIN data repository selection process. We can just refer to that paper for it. 2nd, the clusters are all of different sizes Cluster 1 being 86 patients and Cluster 6 being 11 patients. This makes analysis skewed and may be a little difficult to interpret. This should at least be listed in the limitations section. The outcomes of ICU mortality makes the clinical implications of this study a little difficult. If possible, should include hospital mortality and possibly 30/60 day mortality.

6. PLOS authors have the option to publish the peer review history of their article (what does this mean?). If published, this will include your full peer review and any attached files.

Reviewer #1: No

Reviewer #2: No

---

## [Author Response · Author response to Decision Letter 0]

23 Sep 2022

Reviewer #1: The authors propose an unbiased systems biology approach (PCA) to characterize the various sepsis phenotypes based on biomarkers of inflammation (pro and anti), and representing specific immune responses (e.g., IL-18 for inflammasome). The rationale being to better characterize sepsis phenotypes to guide future clinical trials (as per the abstract) and to elucidate disease mechanisms (as per the Introduction). Clinical characterization was based on PIRO, whereas biological characterization was based on an array of 38 selected different proteins and mRNA transcripts. The analysis yielded 6 distinct sub-groups of sepsis, some with higher mortality, and with distinct molecular markers. There are question remaining:

Concerns:

1. Is PCA the best statistical approach? The reliance on a complete dataset leads to the exclusion of many sepsis cases, including perhaps the most import one (those dying within 3 days).

R: To justify the PCA approach, we added the following : “PCA is a statistical procedure that summarizes the information content in large data tables by means of a smaller set of “summary indices” that can be more easily visualized and analyzed. PCA is considered to be the reference method to identify the unobservable, “latent” factors or dimensions that underlie or structure a set of observed variables (ref 28).” And «These components define the dimensionality of the reduced space and correspond to the underlying latent factors or patterns. The remaining components (not retained) represent the residual variability (measurement error, single marker unrelated to the others).”

Regarding the exclusion of the patients dying within 3 days, we must be more explicit about the choice of the 3-day period to assess the phenotypes, as the method section was too short in the initial version of the paper. We used 3 days because we wanted to include the “certainty”, and not the “suspicion”, of infection in the phenotypical criteria of the potential clusters. Therefore, we had to consider around 72 hours between suspicion and confirmation (or not) of the infection. The rationale to describe clusters of phenotypes was to confirm (or not) that different patterns of endotypes (which are intended to describe pathophysiological pathways) were associated with specific phenotypes. Therefore, we consider that phenotypes were outcomes of our research. 

In the revised version, we have added the following in the method section: 

“we wanted to describe their clinical phenotypes including the “certainty”, and not the “suspicion”, of infection in the phenotypical criteria of the potential clusters. The rationale to describe clusters of phenotypes was to confirm (or not) that different patterns of endotypes (which are intended to describe pathophysiological pathways) were associated with specific phenotypes. As phenotypes were outcomes of our research, we considered that 72 hours was necessary to distinguish suspicion and confirmation of the infection.”

2. Although 33 patients died and were therefore excluded, it would be interesting to describe this special sub-group based on the available, albeit limited data.

R In the revised version, we provide a table describing the patients that were excluded because they were discharged or died before day-3. The table includes the same data that we used to describe the PIRO system in the overall population and is displayed in the supplement appendix (S2 Table).

3. Most of the molecules used to characterize sepsis biology are reasonable. Certain common biomarkers associated clinically with sepsis, such as PMN% or blast% were not included. Please explain.

R: We agree that these biomarkers are routine use in several institution. Unfortunately, they were not collected in our research database, because they were not part of the biomarkers we decided to assess in the initial project. 

4. Why was "viral sepsis" not included? What about fungal sepsis?

R: The goal of our first paper was to separate patients with an infection which requires a treatment from the overall patients with inflammation. This was because the goal of detecting sepsis early is to implement antibiotic in due time. Additionally, bacteria are by far the most frequent pathogens that cause sepsis, beside fungi and viruses. As the ICUs that participated in the study were general ICUs, few patients had virus or fungi as a cause of sepsis, except a couple of cases of Influenza. We decided to select only bacteria for the sake of homogeneity of the cohort, and potentially of the mechanisms of the immune response. 

5. It is said the the data was "transformed" and "reduced" prior to establishing clusters. Please provide more detail such that the approach could be reproduced by others.

R : We added the following in the section method : “Transformation and reduction of data were necessary to obtain a homogeneous scale of independent data.” And “For all considered biomarkers, we determined the maximum value of the blood levels (Cmax) obtained at day 0 and day 1 of inclusion. These values were natural log-transformed (ln(marker+0.0001) to normalize their distribution (convert the skewed distribution of these variables to approximate normality) and further standardized to have 0 mean and unit variance.”

6. Figure 2 is excellent and nicely summarizes the results. However, it would be interesting to consider the minimal data set that could be used for future external validation of these sepsis sub-types.

R: Figure 2 is based on both the endotypes (patterns of biomarkers) and the phenotypes (PIRO classification). Future external validation of our results could be based on a minimal dataset based on both the endotypes and phenotypes we found. This data set could be the following: confirmed infection at day 3: yes/no; blood values of CRP and ferritin; use of RRT and/or vaso-pressive drugs; blood levels of TNF-α, IL-6, and IL-8 (pro-inflammatory); blood levels of IL-1Ra and IL-10 (anti-inflammatory; HLA-DR and CD74 (adaptive profile). This is now proposed at the end of the discussion section.

7. From a practical standpoint it is most useful to establish sepsis phenotypes as early as possible (e.g., 3 days would be too long for an interventional clinical trail). As such, it may be more interesting to know what the initial features of each sub-type are rather than "peak values" of biomarkers measured over 24 hrs.

R: One cannot ascertain the infection before day 2 or 3 in most patients. Therefore, only biomarkers present when the first symptoms of inflammation and the suspicion of sepsis occur may be useful to select patients in future trials, or to adjust causal (anti-bacterial) therapy of sepsis. Therefore, we consider that only the endotypes (and not the phenotypes) that we found may be useful for this purpose. We chose peak values because we were unable to ascertain the precise date of sepsis onset, due to the variability of the syndrome and the high number of biomarkers used, leading to missing values in several biomarkers. In future use of these endotypes, a selection of 2 or 3 biomarkers for each immune pathway would be sufficient, making the use of the first samples more feasible. 

We have added the following in the method section:

“We chose peak values because we were unable to ascertain the precise date of sepsis onset, due to the variability of the syndrome and the high number of biomarkers used, leading to missing values in several biomarkers.”

We have added the following in the discussion section:

“One cannot ascertain the infection before day 2 or 3 in most patients. Therefore, only biomarkers present when the first symptoms of inflammation and the suspicion of sepsis occur may be useful to select patients in future trials, or to adjust causal (anti-bacterial) therapy of sepsis. Therefore, we consider that only the endotypes (and not the phenotypes) that we found may be useful for this purpose. In future use of these endotypes, a selection of 2 or 3 biomarkers for each immune pathway would be sufficient, limiting the risk of missing values at the first time point.”

Reviewer #2: This study was interesting in clustering sepsis profiles and analyzing them across various clinical and management factors and bio markers. As this was a secondary data analysis, not sure why a significant portion of the methods section was used to describe the CAPTAIN data repository selection process. We can just refer to that paper for it. 

R: In the revised version, we refer to the paper we published previously for a larger part of the biomarker assessment techniques and data management. A part of the method section is also provided in the supplementary appendix.

2nd, the clusters are all of different sizes Cluster 1 being 86 patients and Cluster 6 being 11 patients. This makes analysis skewed and may be a little difficult to interpret. This should at least be listed in the limitations section. 

R: PCA is a technique that separates patients into groups of homogeneous characteristics. In this study, the characteristics are biomarker patterns. The number of components does not depend on the percentage of variance to explain. Components that explain 5 percent of the variance may not be insignificant. Several groups are small but present very distinctive phenotypic patterns by comparison to the other ones. Therefore, we believe their description is interesting, as some searchers may want to work on these specific patterns. 

This has been better specified in the method section, as follows: “The number of components-patterns to retain was determined by the Horn and Velicer methods as recommended [ref 27]. These components define the dimensionality of the reduced space and correspond to the underlying latent factors or patterns. The remaining components (not retained) represent the residual variability (measurement error, single marker unrelated to the others).”

We also have added the following sentences in the limitations paragraph of the discussion section : « Third, the limitations of the statistical methods should be borne in mind. It is useful to recall that the patterns and clusters that emerge from factorial and taxonomic methods do not exactly correspond to clear-cut groups or endotypes. Despite following recommendations for optimizing method implementation and enhancing reliability of results, emerged clusters may be polluted by misclassification of statistical nature and their meaning require careful analysis. It is therefore crucial to check the stability of the clusters obtained, especially of the smaller ones.” 

The outcomes of ICU mortality makes the clinical implications of this study a little difficult. If possible, should include hospital mortality and possibly 30/60 day mortality.

R: In the revised version, we provide the mortality at day-14, day-30, and day-60 in table 2, instead of the ICU mortality.

---

## [Decision Letter · Decision Letter 1]

11 Oct 2022

Clustering ICU patients with sepsis based on the patterns of their circulating biomarkers: a secondary analysis of the CAPTAIN prospective multicenter cohort study.

PONE-D-22-10420R1

Dear Dr. Misset,

We’re pleased to inform you that your manuscript has been judged scientifically suitable for publication and will be formally accepted for publication once it meets all outstanding technical requirements.

Kind regards,

Florian Uhle

Academic Editor

PLOS ONE

Additional Editor Comments (optional):

Reviewers' comments:

Reviewer's Responses to Questions

**Comments to the Author**

1. If the authors have adequately addressed your comments raised in a previous round of review and you feel that this manuscript is now acceptable for publication, you may indicate that here to bypass the “Comments to the Author” section, enter your conflict of interest statement in the “Confidential to Editor” section, and submit your "Accept" recommendation.

Reviewer #1: All comments have been addressed

Reviewer #2: All comments have been addressed

2. Is the manuscript technically sound, and do the data support the conclusions?

Reviewer #1: Yes

Reviewer #2: Yes

3. Has the statistical analysis been performed appropriately and rigorously? 

Reviewer #1: Yes

Reviewer #2: Yes

4. Have the authors made all data underlying the findings in their manuscript fully available?

Reviewer #1: Yes

Reviewer #2: No

5. Is the manuscript presented in an intelligible fashion and written in standard English?

Reviewer #1: Yes

Reviewer #2: Yes

6. Review Comments to the Author

Reviewer #1: Early endotyping of bacterial sepsis could aid in the design of future clinical trials and may also guide certain clinical interventions (e.g., palliative care discussions). This work also has implications for the design future mechanistic studies to better understand the relations between biomarkers and outcomes.

Reviewer #2: Thank you for addressing the questions. I am interested to see further long term outcomes on subsequent papers.

7. PLOS authors have the option to publish the peer review history of their article (what does this mean?). If published, this will include your full peer review and any attached files.

Reviewer #1: No

Reviewer #2: No

---

## [Editor Report · Acceptance letter]

18 Oct 2022

PONE-D-22-10420R1 

Clustering ICU patients with sepsis based on the patterns of their circulating biomarkers: a secondary analysis of the CAPTAIN prospective multicenter cohort study. 

Dear Dr. Misset:

I'm pleased to inform you that your manuscript has been deemed suitable for publication in PLOS ONE. Congratulations! Your manuscript is now with our production department. 

Kind regards, 

on behalf of

Dr. Florian Uhle 

Academic Editor

PLOS ONE